# Effect of the Age and Body Weight of the Broiler Breeders Male on the Presentation of Oxidative Stress and Its Correlation with the Quality of Testicular Parenchyma and Physiological Antioxidant Levels

**DOI:** 10.3390/vetsci7020069

**Published:** 2020-05-26

**Authors:** Magdalena Escorcia, Félix Sánchez-Godoy, David Ramos-Vidales, Omar Noel Medina-Campos, José Pedraza-Chaverri

**Affiliations:** 1Departamento de Medicina y Zootecnia de Aves, Facultad de Medicina Veterinaria y Zootecnia-UNAM, Av. Universidad 3000, Colonia, C.U., Coyoacán CDMX 04510, Mexico; spuma91@hotmail.com; 2Centro de Enseñanza, Investigación y Extensión en Producción Avícola CEIEPAv-UNAM, Tláhuac, D.F. Manuel M. López s/n, Santa Ana Poniente, Tláhuac CDMX 13300, Mexico; mvz.ramosvd@outlook.es; 3Departamento de Biología, Facultad de Química-UNAM, Av. Universidad 3000, Col. UNAM, CU., Coyoacán CDMX 04510, Mexico; omarnoelmedina@gmail.com (O.N.M.-C.); pedraza@unam.mx (J.P.-C.)

**Keywords:** antioxidant capacity, oxidative stress, fertility, broiler breeder male, testicles

## Abstract

Chicken meat is a food of high nutritional quality; its production requires birds called broilers breeders and looking after all aspects that influence their reproductive capacity. An ongoing controversy exists among researchers related to the weight of the rooster and its fertilization capacity. By histological and biochemical tests, the association between weight and age with oxidant damage, testicular parenchyma and antioxidant capacity was evaluated in Ross 308 roosters. Testicular integrity was assessed by histological analysis, oxidative stress was determined by malondialdehyde content, non-enzymatic antioxidant capacity was determined by oxygen radical absorbance capacity assay and enzymatic antioxidant capacity through glutathione peroxidase, glutathione reductase and glutathione-S-transferase activities. Histological analysis showed vacuolization of the epithelium from the seminiferous tubules. A significant negative association was observed between malondialdehyde and the deterioration of the integrity of the seminiferous epithelium, as well as between age and integrity of the seminiferous epithelium. It became evident that oxidative damage directly affects the quality of testicular parenchyma. Weight and age were not associated with the antioxidant enzymes activities, but with non-enzymatic capacity. The data obtained suggest that weight is not the most important factor that influences the fertility of the rooster.

## 1. Introduction

The male breeding broiler is the bird used to produce a fertile egg from which the future broiler will be fattened to obtain meat. The importance of the rooster lies in the dissemination of genetic progress to obtain, in its progeny, large muscle masses because during its reproductive stage it is responsible for the fertilization of 1000 to 2000 eggs [1].

Currently, due to the genetic improvements that have been carried out to obtain birds with large muscle masses, breeding roster weight gain is easily achieved. The breast is the body part that reaches the greatest weight and volume [2], however, this weight gain contrasts with a decrease in reproductive rates [3].

Regarding age, the reproductive rooster begins to ejaculate at the 25th week, the maximum fertility rate is reached around 40 weeks and decreases significantly around the 45th week [1,3]. The rooster reproductive system consists of two testicles, with their respective epididymis, vas deferens and an intromittent organ [4,5]. The testicles develop their maximum potential after 29 weeks of age, observing an increase in the tubular diameter and in the production of sperm that move freely in the light of the seminiferous tubules. During adult age (30–50 weeks), testicular functional integrity is accomplished. Likewise, at the end of this period, the appearance of interstitial fibrosis, testicular atrophy, and decreased sperm production are observed. These changes are more evident as the rooster ages and at week 60 there is a marked decrease in the number of sperm in the light of the tubules, as well as the presence of moderate atrophy, sperm degeneration and calcification of the seminiferous tubules [6].

The integrity and function of the testicular parenchyma are important for the luteinizing hormone to activate Leydig cells and stimulate the secretion of testosterone, which is the hormone that triggers the sexual appetite of the rooster [7] as well as for the follicle-stimulating hormone to favor the proliferation of Sertoli cells through the production of cyclic adenosine monophosphate (cAMP) [8]. Sertoli cells are somatic cells that nourish and protect the germ cell populations throughout spermatogenesis. These cells are highly sensitive to external factors and have a high metabolism to provide nutrients and energy to the germ cells [8,9,10].

These physiological processes implies a significant energy demand that is obtained from the mitochondria that produces energy in form of adenosine triphosphate through oxidative phosphorylation, a process that involves the electron transport to the oxygen.

Oxygen is essential for aerobic organisms, but it also generates reactive oxygen species (ROS), which are molecules that are originated from the sequential addition of electrons to the oxygen and therefore have a higher reactivity. Among these reactive species can be found free radicals such as superoxide molecules (O_2_•−), hydroxyl radicals (HO•), reactive nitrogen species [derivate from nitric oxide (NO)] and peroxides, both organic and inorganic (i.e., hydrogen peroxide (H_2_O_2_) and hypochlorous acid (HOCl)) [11]. 

ROS play an important role in the signaling pathways of the metabolic processes; however, when they are present in excessive levels, they can cause noxious effects on cells such as lipid peroxidation, deoxyribonucleic acid (DNA) alteration or protein degradation [11,12]. To avoid such effects, organisms have antioxidant systems that are classified as enzymatic and non-enzymatic. The antioxidant enzymes catalyze chemical reactions using substrates that, simultaneously, chemically react with ROS [11]. Examples of these are the superoxide dismutase (SOD), catalase (CAT), glutathione peroxidase (GPx), and glutathione reductase (GR). However, even in the presence of antioxidant systems ROS can exceed the antioxidant capacity and lead the establishment of a condition known as oxidative stress [13].

Physiological levels of ROS, produced by the mitochondrial respiratory chain, are necessary for normal sperm behavior, including capacitation, hyperactivation, acrosome reaction, egg fusion, and fertilization but when sperm is exposed to oxidative stress a poor sperm quality is observed [14]. De Lamirande and Gagnon [15] demonstrated that oxidative stress modifies the cytoskeleton and sperm axoneme causing a reduction in sperm motility. On the other hand, Aitken et al. [16] showed that oxidative stress-induced damage to the plasmatic membrane of sperm affects the fusion between sperms and oocytes. In the case of birds, Eid et al. [17] demonstrated that oxidative stress negatively influences several aspects of reproduction, such as spermatogenesis, the quantity, and quality of fertile egg and the offspring viability [15,16,17].

In general, studies to determine the infertility in men and animals have been aimed at establishing oxidative stress damage in sperm, however, to date, there are no studies that determine the damage that the oxidative stress can cause to the testicular parenchyma. Based on the foregoing, the aim of this study was to assess whether there is an association between age and weight with oxidative damage to the testis and the antioxidant capacity of this organ.

## 2. Materials and Methods

Animals used in this study were obtained from a broiler breeders farm and were handled according to the dispositions of the official Mexican norm NOM-062-ZOO-1999 [18]. The Project was approved with the number 56 by the Internal Committee for the Care and Use of Laboratory Animals (CICUA) from the school of Veterinary Medicine and Zoo Technology from the National Autonomous University of Mexico.

### 2.1. Animals

The sample consisted of three batches of Ross 308 roosters, randomly obtained from a commercial farm. Each group consisted of 12 animals according to their age, as follows. Group (A) 30 weeks old, group (B) 36 weeks old and group (C) 62 weeks old. Before the sacrifice of the birds, the body weight of each rooster was obtained.

### 2.2. A Sampling of Testis for Histologic Interpretation and Oxidative Stress Assessment

All animals studied were sacrificed using sodium thiopental intravenously at a dose of 4 mg/kg live weight. Testis were gently dissected and removed from their original anatomical position; subsequently they were washed with physiological saline solution to remove excess blood. A 0.5 cm thick cut was obtained from the equator of each right testicle and each section was prepared for the histological analysis by placing them in a 10% buffered formalin solution followed by paraffin embedding. On the other hand, left testicles were immediately placed on liquid nitrogen and stored at −80 °C until used for the corresponding biochemical determinations.

### 2.3. Histological Analysis of the Testicular Parenchyma

This analysis was performed on the testicular tissue of all sampled roosters. Subsequently, from paraffin embedding, slices of 3 μm in thickness were obtained, processed and stained with the hematoxylin-eosin (H&E) technique according to the protocol established by Prophet et al. [19].

### 2.4. Description of the Histologic Assessment

Each histological section stained with H&E and five random fields were evaluated under 100× magnification. To estimate the testicular integrity of each group of rosters, the seminiferous epithelium thickness, the degree of sperm maturation, and visual evaluation of the size of the seminiferous tubules were assessed categorizing the morphological changes in three grades: grade 3, no changes; grade 2, moderate changes; and grade 1, severe changes.

### 2.5. Oxidative Stress and Antioxidant Capacity in Testicular Tissue

The testicle samples were homogenized in 75 mM phosphate buffer, pH 7.4 (1:2, *w*/*v*) and centrifuged at 9000× *g* at 4 °C for 10 min. The supernatants were stored at −80 °C until the aforementioned determinations were carried out.

#### 2.5.1. Malondialdehyde (MDA) Content

MDA content was determined as follows: A mixture of 0.2 mL of supernatant, 0.65 mL 10 mM 1-methyl-2-phenylindole in acetonitrile/methanol (3:1 *v*/*v*) and 0.15 mL concentrated HCl was incubated at 45 °C for 40 min and centrifuged at 3000× *g* for 5 min. The supernatant absorbance at 586 nm was interpolated in a tetramethoxypropane standard curve and results were expressed as nmol MDA/mg protein [20].

#### 2.5.2. Non-Enzymatic Antioxidant Capacity

This parameter was evaluated using oxygen radical absorbance capacity (ORAC) assay according to the protocol established by Huang et al. [21]. 2,2′-Azobis(2-methylpropionamidine) dihydrochloride (AAPH) was used as a peroxyl radical generator, Trolox was used as standard and fluorescein was used as a fluorescent probe. Briefly, the fluorescence of a mixture of 25 μL of water, Trolox standard, diluted samples or diluted vehicle, 25 μL of 153 mM AAPH and 150 μL of 50 nM fluorescein was measured every minute for 90 min at 37 °C using an excitation wavelength of 485 nm and an emission wavelength of 520 nm. The ORAC values were calculated using the net area under the decay curves and were expressed as μmoles of Trolox equivalents (TE)/mg protein.

#### 2.5.3. Antioxidant Enzyme Activity Assays

##### Glutathione Peroxidase (GPx)

GPx activity was measured according to Lawrence and Burk [22]. One-hundred μL of supernatant was added to 0.8 mL of mixture reaction (1 mM EDTA, 1 mM NaN_3_, 0.2 mM NADPH, 1 U/mL of the GR and 1 mM glutathione in 50 mM potassium phosphate buffer pH 7.0) and 0.1 mL 2.5 mM H_2_O_2_. Optical density at 340 nm was recorded each min for 3 min and the activity was calculated from the slope using the extinction coefficient of NADPH at 340 nm (6.22 mM^−1^ cm^−1^). Data were expressed as units (μmoles of NADPH oxidized/min)/mg protein.

##### Glutathione Reductase (GR)

GR activity was determined according to Carlberg and Mannervik [23] as follows: 0.05 mL of supernatant was added to 0.95 mL of mixture reaction (1.25 mM oxidized glutathione, 0.5 mM EDTA and 0.1 mM NADPH in 100 mM potassium phosphate buffer pH 7.6). The activity determination was done in the same way as for GPx [22,23].

##### Glutathione -S-Transferase (GST)

The activity was determined using the method of Habig et al. [24]. Briefly, 0.02 mL of supernatant was added to 0.98 mL of reaction (2 mM reduced glutathione, 1 mM 1-chloro-2,4-dinitrobenzene (CDNB) in 50 mM potassium phosphate buffer pH 6.5). The absorbance at 340 nm was recorded for 3 min and activity was expressed as µmoles of CDNB conjugate formed/min/mg protein using a molar extinction coefficient of 9.6 mM^−1^ cm^−1^.

### 2.6. Total Protein Determination

This was carried out using the method of Lowry et al. [25].

### 2.7. Statistical Analysis

The three age groups were compared with one single way single factor analysis of variance (ANOVA). For variables with significant statistical difference, the Tukey test was applied for comparison between the means with a level of significance α = 0.05.

Correlation analyses were performed using Pearson’s methodology in order to determine if there is an association between the body weight of the roosters with (a) the structural integrity of the testicular tissue, (b) the damage affected by oxidative stress, (c) non-enzymatic antioxidant capacity (ORAC) and (d) the enzymatic antioxidant activities.

For all statistical analyzes referring to weight, standard weight provided by Ross 308 breeder house (Table 1). The data processing and analysis were performed with the statistical package R [26].

## 3. Results

### 3.1. Corporal Weight, Age, and Integrity of Testicular Parenchyma

Visual evaluation of the histological sections of the testicular parenchyma from the three age groups revealed the presence of vacuoles in the epithelium of the seminiferous tubules at different degrees of severity, regardless of the age of the rooster (Figure 1 and Figure 2).

The variables evaluated to determine the integrity of testicular parenchyma were (a) visual evaluation of the size of the seminiferous tubules, (b) degree of sperm maturation, and (c) seminiferous epithelium thickness.

#### 3.1.1. Histological Analysis of the Testicular Parenchyma

Figure 3 shows the distribution of the histological finding counts of: Figure 3A the seminiferous tubules, Figure 3B the sperm maturation, Figure 3C the integrity of the seminiferous epithelium and Figure 3D the seminiferous epithelium thickness per age.

The size of the seminiferous tubules and the maturity of the sperm presented a higher normal score count in the three evaluated ages while the integrity of the seminiferous tubules at week 30 of age presented only normal findings. However, at weeks 32 and 62 higher counts of moderate to severe changes were observed. Regarding to the seminiferous epithelium thickness it was observed that alterations become more severe as the age of rooster increases. No association was found between the age of the rosters with the size of the seminiferous tubules (*p* = 0.240), sperm maturity (*p* = 0.105) or seminiferous epithelium thickness (*p* = 0.108).

The results show that there is an association between the integrity of the seminiferous tubules and the age of the roosters (*p* = 0.0007), which confirms the tendency seen in the Figure 3C. The older roosters (36 and 62 weeks of age) showed moderate changes in the integrity, unlike the younger roosters that did not show alterations.

#### 3.1.2. Integrity of the Seminiferous Epithelium

According to the results obtained, there is no correlation between the corporal weight and the variables evaluated to determinate the integrity of testicular parenchyma. However, there was a negative significant association between age and maturity of the sperms (*p* = −0.34, *p* = 0.043), as well as between age and size of the seminiferous tubules (*p* = −0.47, *p* = 0.004), and between age and seminiferous epithelium thickness (*p* = −0.33, *p* = 0.048). On the other hand, although the size of the seminiferous tubes decreases as the age of the rooster increases, the association between these last two variables was not significant (*p* = −0.26, *p* < 0.13).

### 3.2. Correlation of Body Weight, Age with Oxidative Stress and Integrity of Testicular Parenchyma

Regardless of age and weight, there was a significant negative association between MDA content and the integrity of seminiferous epithelium (*p* = −0.46, *p* = 0.0047).

### 3.3. Biochemical Correlations

The results of the correlation between the weight of roosters at different ages with oxidative tissue damage (MDA), non-enzymatic antioxidant capacity (ORAC) and enzymatic antioxidant capacity (GPx, GR, and GST) are shown in Table 2. It was observed a positive correlation between the weight of the roosters of 30 weeks of age and MDA content (*r* = 0.760, *p* = 0.029). No association was found in any other age between the weight of the roosters and the variables evaluated (*p* > 0.05).

Table 3 presents the correlations matrix between oxidative stress (MDA), non-enzymatic antioxidant capacity (ORAC) and enzymatic antioxidant capacity (GPx, GR, and GST). MDA content and ORAC showed a positive and highly significant association regardless the age and weight (*r* = 0.528, *p* = 0.002). It was not found a correlation between MDA and antioxidant enzymes: MDA with GPx (*r* = −0.320), MDA with GR (*r* = 0.180) and MDA with GST (*r* = 0.130). The association between ORAC and GPx was not significant (*r* = 0.191, *p* = 0.265) but a highly positive association (*p* < 0.001) was found for ORAC with GR (*r* = 0.635), and ORAC with GST (*r* = 0.595).

Finally, a highly positive significant association was detected (*p* < 0.001) between the three antioxidant enzymes. The correlation values for GR and GPx, GPx and GST, and GST and GR were 0.674, 0.658 and 0.843, respectively.

### 3.4. Age Group Comparisons: Oxidative Stress and Antioxidant Capacity

The roosters were grouped by age and subsequently compared in terms of the results obtained in the biochemical measurements. Significant statistical differences (*p* < 0.05) were found between the age groups for MDA and ORAC. The roosters of 30 weeks of age had a lower MDA content compared to the roosters of 36 and 62 weeks of age, while the animals of 36 and 62 weeks of age had similar levels of MDA. Similar results were found in ORAC since roosters of 30 weeks old presented a lower non-enzymatic antioxidant capacity compared to the rosters of 36 and 62 weeks of age, while rosters of 36 and 62 weeks of age presented similar levels of ORAC (Figure 4). No significant statistical differences were found between the age groups for the GPx, GR and GST activities; the results are shown in Figure 5.

## 4. Discussion

From the histological point of view, one of the main lesions observed in all testicular samples obtained from the three age groups was the presence of vacuoles in the epithelium of the seminiferous tubules at different degrees of severity. Given the observations of Sarabia et al. [6], Zitzmann et al. [27], and Rodriguez et al. [28], we suggest that apoptosis could be the cause of the origin of the vacuoles observed in Sertoli cells and Leydig cells [6,27,28]. Then the presence of vacuoles observed since 30 weeks of age may be associated to the fact that after birth and during development some cells suffer apoptosis and that the testicular modeling is present throughout the life of the rooster [28]. It would be interesting to determine if the presence of vacuoles occurs before 30 weeks of age, from the moment the rooster reaches sexual maturity.

It is important to note that despite the relationship between the reduction of testis size and the end of the breeding seasons of different bird species, there are not enough studies that have related apoptosis in the testicular parenchyma of these birds and the formation of the vacuoles cited, nor has it been established whether the vacuoles are the consequence of the reduction in the size of the testicles.

Alonso-Alvarez et al. [29] suggest that reproductive effort in species, in general, can profoundly influence susceptibility to oxidative stress. According to the manuals of the genetic house of the roosters with which we work, it is around 30 weeks of age when this type of birds are at their peak of production of mature sperm cells [3], which leads to a greater energy demand causing generation of ROS. Coinciding with the above, Wilson et al. [30], report a rapid reduction of Sertoli cells number between 22 and 24 weeks of age, which are displaced by the appearance of spermatogenesis.

Data related to fertility and body weight provides confusing information, Hocking [31] suggested that fertility in roosters could be higher in animals with weights higher than the standard, which contrasts with the report of Renema et al. [32] that indicates that weights higher than standard values influence reproductive capacity. Contrary to expectations, and despite the fact that the oxidative damage was greater in the testicles of roosters of 36 and 62 weeks old, when compared with the testicles of roosters of 30 weeks of age (Figure 3), our data do not show a correlation between the results obtained and the weight of the animal (Table 2), a situation that suggests that the weight does not influence the reproductive capacity of the roosters evaluated [31,32]. Wilson et al. [30] reported that the number of Sertoli cells in commercial roosters increased significantly from week 1 to week 21 post-hatching, but subsequently there were no changes until week 64. Then the absence of changes in the number of Sertoli cells in each age period could be the cause of the lack of statistical difference in the quantification of MDA between the roosters of 36 and 62 weeks of age.

According to a literature review about histomorphological studies in elderly men conducted by Gunes et al. [33], a decrease in the number of germ cells and Sertoli cells attributable to age was demonstrated. The same authors found that in the course of aging, the thickness of the basal membrane of the seminiferous tubules increased, while the epithelium of the seminiferous tubules decreased, a situation that in our study with roosters was also observed. Coinciding with the above, Paniagua [34] described the onset of tubular involution due to age, which results in testicular atrophy. Based on the results obtained in this work and based on what is quoted by Gunes et al. [33] and Paniagua [34], it can be affirmed that it is age, and not weight, that influences the cock’s fertility capacity.

According to our results, there is no correlation between the weight of the rooster and the enzymatic antioxidant capacity, however, the behavior of the non-enzymatic antioxidant capacity (ORAC) was graphically very similar to the behavior of MDA content, a situation that is confirmed with the results obtained through the Pearson methodology in which the correlation between MDA and ORAC (Unpublished data) is evidenced. This could be interpreted as the body’s response to try to counteract oxidative stress, despite the lack of association between this variable and the body weight of the bird.

Concerning the enzymatic antioxidant response, we must remember that birds in general, compared to other vertebrate animals, including humans, have a higher metabolic rate, higher blood glucose levels, a higher body temperature and a higher pressure partial pulmonary oxygen. Although these physiological conditions predispose to a greater generation of ROS, it has been observed that birds are resistant to the damage that this condition predisposes. Monnier [35] proposed that birds should have mechanisms that prevent such damage. Gutierrez et al. [36] found that the enzymatic antioxidant activity remained unchanged during periods of metabolic demand of birds such as migration, suggesting that birds improve the oxidative state during migratory preparation, which could represent an adaptation to reduce the physiological costs of long-distance migration. Based on the above, and until proven otherwise, we suggest that the behavior of the enzymatic antioxidant capacity described in the present work could be a consequence of the adaptation of the roosters to the metabolic requirements during their reproductive period [35,36].

Despite the previous statement and regardless of the weight of the roosters, there is a positive correlation between the enzymes evaluated. The above coincides with the studies of Jedrzejczak et al. [37] and Kasimanickam et al. [38], who demonstrated that the GPx enzyme is key in maintaining the pro and antioxidant balance, since it eliminates peroxides from cells, turning them into non-reactive products. Related to this detoxification reaction Partyka et al. [39] pointed out that oxidized glutathione formed by GPx activity is reduced by the GR enzyme to regenerate the reduced glutathione required for GPx. This initiates a chain reaction that, depending on the stimulus, involves different antioxidant enzymes, likewise, this could be the explanation of the correlation obtained in the present work between enzymatic and non-enzymatic antioxidant capacity [36,37,39].

## 5. Conclusions

The consistency between the lack of correlation between the weight of the rooster and the variables evaluated gives us the elements to conclude that weight is not the most important variable that influences oxidative stress, testicular parenchyma architecture, enzymatic and non-enzymatic antioxidant capacity, with this conclusion we must not forget that the biological processes of spermatogenesis can be affected by different factors that cause alterations in the reproductive performance of the rooster.

## Figures and Tables

**Figure 1 vetsci-07-00069-f001:**
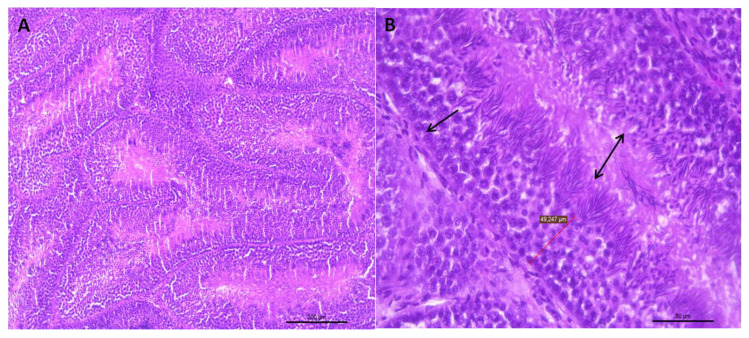
(**A**) Representative transverse section of seminiferous tubules of the testes of roosters from 30 weeks of age with standard weight and a value of 3 for testicular integrity. Small dark bodies are observed that correspond to different stages of sperm maturation. HE staining (bar: 200 μm). (**B**) Area extension of photograph A, where the epithelium of the seminiferous tubule is shown (arrow). This epithelium is formed by germ cells, which, as they move towards the tubule lumen, can be found in different spermatogenesis stages. Towards the tubule lumen, mature sperm clusters can be seen in which the head and the tail can be differentiated (double-headed arrow). HE staining (bar: 50 μm).

**Figure 2 vetsci-07-00069-f002:**
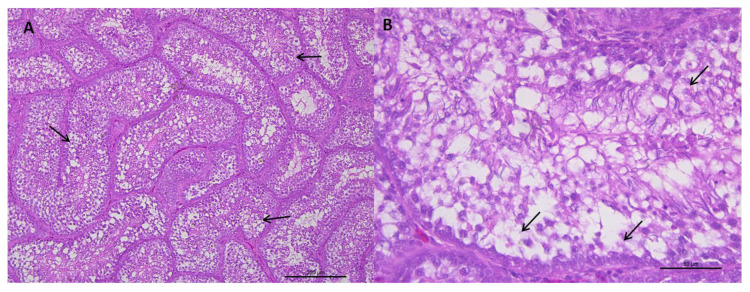
(**A**) Representative transverse section of seminiferous tubules of roosters from 62 weeks of age with weights below the standard and a value of 1 for testicular integrity. Multifocal vacuoles (arrows) are observed in the epithelium of the seminiferous tubules and a severe depopulation of the spermatic line. HE staining (bar: 200 μm). (**B**) Area extension of photograph A. Vacuolization and disorganization of the seminiferous tubule epithelium is observed, as well as, cell depopulation of the spermatic line (arrows). HE staining (bar: 50 μm).

**Figure 3 vetsci-07-00069-f003:**
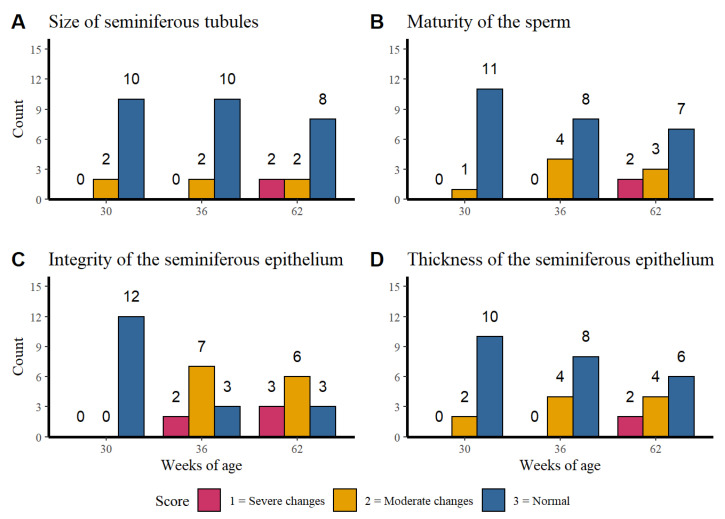
Distribution of the histological finding counts of the seminiferous tubules (**A**), sperm maturation (**B**), integrity of the seminiferous epithelium (**C**), and seminiferous epithelium thickness (**D**) per week of age.

**Figure 4 vetsci-07-00069-f004:**
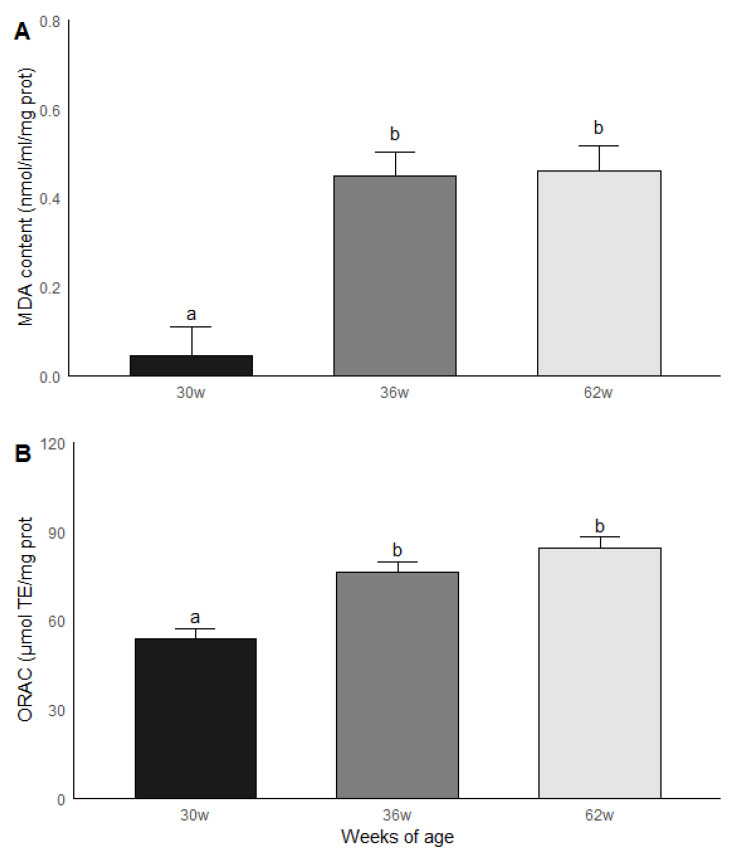
Malondialdehyde (MDA) (**A**) content and (**B**) non-enzymatic antioxidant capacity (ORAC) in testicular parenchima of roosters of 30, 36 and 62 weeks of age. Data are mean ± SEM of measurements by duplicate. The roosters of 30 weeks of age had a lower MDA content compared to the roosters of 36 and 62 weeks of age, while the animals of 36 and 62 weeks of age had similar levels of MDA. (**B**) Similar results were found in ORAC since roosters of 30 weeks old presented a lower non-enzymatic antioxidant capacity compared to the rosters of 36 and 62 weeks of age, while rosters of 36 and 62 weeks of age presented similar levels of ORAC. Data are mean ± SEM of measurements by duplicate. ^a, b^ means with different letters show significant difference (*p* < 0.05).

**Figure 5 vetsci-07-00069-f005:**
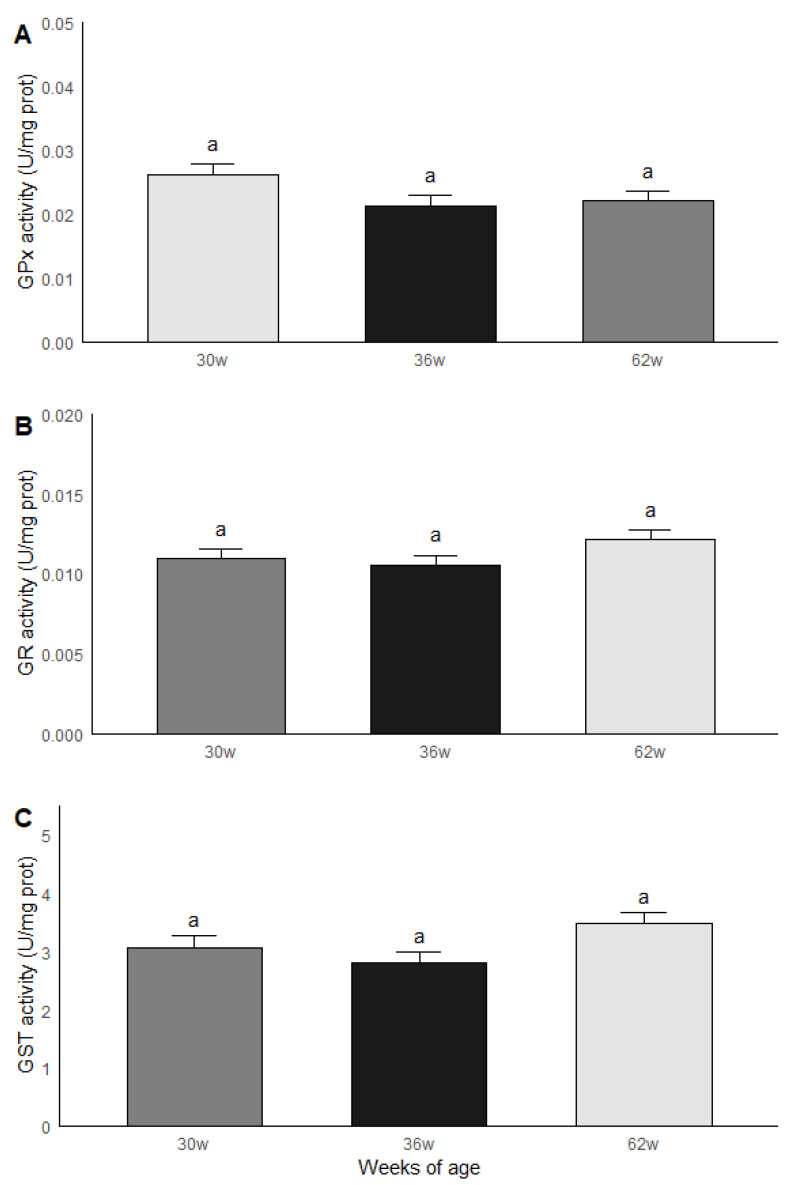
GPx (**A**), GR (**B**) and GST (**C**) activities in testicular parenchima of roosters af 30, 36 and 62 weeks of age. ^a^ Equal literals - no significant statistical difference were found between the age groups for the GPx, GR and GST activities. Data are mean ± SEM of measurements by duplicate.

**Table 1 vetsci-07-00069-t001:** Standard weight of breeder broiler rooster.

**Age (Weeks)**	30	36	60
**Weight (kg)**	4.150	4.330	5.050

Taken from the Aviagen genetic house management manual [3].

**Table 2 vetsci-07-00069-t002:** Correlations between body weight with oxidative stress, non-enzymatic antioxidant capacity and enzymatic antioxidant activities in roosters of different ages.

Variable Evaluated	Weeks of Age					
	30		36		62	
	*r*	*p*	*r*	*p*	*r*	*p*
MDA	0.760	0.029 *	−0.246	0.442	−0.370	0.237
ORAC	−0.200	0.532	0.154	0.632	0.017	0.959
GPx	−0.117	0.718	0.125	0.699	0.138	0.669
GR	−0.252	0.429	0.160	0.620	−0.015	0.963
GST	−0.320	0.311	−0.002	0.995	−0.050	0.877

MDA: Malondialdehyde, ORAC: Oxygen radical absorbance capacity, GPx: Glutathione peroxidase, GR: Glutathione reductase, GST: Glutathione-S-transferase. *r* = Pearson correlation coefficient, *p* = probability value. * significant.

**Table 3 vetsci-07-00069-t003:** Matrix of correlations between oxidative tissue damage, non-enzymatic antioxidant capacity and enzymatic antioxidant capacity.

Variable Evaluated	Probabilistic Variable	MDA	ORAC	GPx	GR	GST
MDA	*r*	1				
	*p*					
ORAC	*r*	0.528	1			
	*p*	0.002 **				
GPx	*r*	−0.320	0.191	1		
	*p*	0.074	0.265			
GR	*r*	0.180	0.635	0.674	1	
	*p*	0.325	<0.001 **	<0.001 **		
GST	*r*	0.130	0.595	0.658	0.843	1
	*p*	0.478	<0.001 **	<0.001 **	<0.001 **	

MDA: Malondialdehyde, ORAC: Oxygen radical absorbance capacity, GPx: Glutathione peroxidase, GR: Glutathione reductase, GST: Glutathione-S-transferase. *r* = Pearson correlation coefficient, *p* = probability value, ** highly significant.

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
