# Peer review of "Effect of the Age and Body Weight of the Broiler Breeders Male on the Presentation of Oxidative Stress and Its Correlation with the Quality of Testicular Parenchyma and Physiological Antioxidant Levels"

_vetsci, 2020, doi:10.3390/vetsci7020069_

Round 1

Reviewer 1 Report

Despite the theme of the present review paper is within the overall scope of the journal and the work deals a potential interesting topic, the manuscript was not adequately presented. All sections need to be revised. Further, the Authors failed to present the paper according to journal’s guidelines in some sections. Moreover, and I think it is the main issue, the English should be revised and improved to make this paper suitable for an international journal. It is quite hard to understand some sentences... Thus, all issues merit more attention, modification and/or removal.

Leave out these reasons, I think it is "useless" to list the specific inaccuracies contained in the manuscript. Therefore, in my opinion, the present manuscript has a lot of negative aspects that preclude its acceptance in Veterinary Sciences. After having reviewed this manuscript, I have failed to find ways in which this manuscript can be improved so that it can be considered for publication.

Reviewer 2 Report

the manuscript is well written. The topic fall within the general scope of the Animals Journal, is innovative and the study adds interesting information in the field of the rooster fertility. It is acceptable for pubblication in its present form.

Round 2

Reviewer 1 Report

The Authors have improved their manuscript, so this revised version merits the final acceptance.